# Forecasting drug resistant HIV protease evolution

Manu Aggarwal[ID]*, Vipul Periwal

National Institutes of Health, Bethesda, Maryland, United States of America

* manu.aggarwal@nih.gov

## Abstract

Protease inhibitors (PIs) target the protease (PR) enzyme to suppress viral replication. Their efficacy in human immunodeficiency virus treatment is compromised by the emergence of drug-resistant strains. Therefore, forecasting drug-resistance during viral evolution would help in the design of effective treatment strategies. To this end, we develop a framework that bridges two distinct data sets. First, we train probabilistic models to learn coevolutionary information in observed PR genotypes in different PI treatment regimens. We use these models to infer transition probabilities of point-mutations conditioned on the genotype and the treatment regimen. Second, we train another set of models to infer drug resistance of PR genotypes to different PIs using data of clinically measured drug resistance. We use these models together to simulate evolutionary trajectories and predict drug resistance. Importantly, we use these simulations to forecast the emergence of persistent drug resistant genotypes. Our analysis shows that the dual therapy of Atazanavir (ATV) and Ritonavir (RTV) is the multi-PI treatment regimen least likely to induce drug resistance. We also conduct an exhaustive ablation study of all possible mutations and predict seven point-mutations as critical for drug resistance. Interestingly, our results highlight the necessity of the amino-acid polymorphism of L63P by predicting that it is critical in developing resistance to Nelfinavir (NFV). The results validate that our framework effectively extracts and combines biological information from the distinct data sets of observed genotypes and drug resistance, while also tackling the challenge of sparsity of available sequence data compared to the large combinatorial complexity of protein evolution and changing functionality in dynamic environments.

## Author summary

The human immunodeficiency virus (HIV) rapidly evolves to evade medication, leading to drug resistance—a major global health challenge. Predicting the evolutionary paths the virus might take under different drug treatments could help us stay one step ahead. In our study, we developed a computational framework

**Data availability statement:** All data is taken from the publicly available Stanford HIV database [16,17]. The consensus subtype B protease (PR) sequence is taken from https://hivdb.stanford.edu/pages/documentPage/consensus_amino_acid_sequences.html. Code is available at https://github.com/nihcompmed/HIV-evolution.

**Funding:** This research was supported by the Intramural Research Program of the National Institute of Diabetes and Digestive and Kidney Diseases (NIDDK) within the National Institutes of Health (NIH) (ZIA DK075091-12 to VP). The contributions of the NIH author(s) were made as part of their official duties as NIH federal employees, are in compliance with agency policy requirements, and are considered Works of the United States Government. However, the findings and conclusions presented in this paper are those of the author(s) and do not necessarily reflect the views of the NIH or the U.S. Department of Health and Human Services. The funders had no role in study design, data collection and analysis, decision to publish, or preparation of the manuscript. VP and MA received salary from NIH.

**Competing interests:** The authors have declared that no competing interests exist.

to forecast the emergence of drug resistance during evolution of HIV protease, a key protein targeted by many drugs. We used machine learning to learn the 'rules of evolution' from the virus's genetic sequences under different drug environments and combined this with a model that predicts drug resistance. By simulating thousands of evolutionary pathways, our framework identifies the most critical mutations for causing resistance. Our findings confirmed the importance of mutations already known to be problematic in the clinic and also highlighted more subtle interactions, such as a key supporting mutation required for resistance to the drug Nelfinavir. Our work offers a new way to computationally explore viral evolution, providing insights that could help design more durable treatment strategies and next-generation drugs to combat HIV.

## 1 Introduction

Human immunodeficiency virus (HIV) antiretroviral therapy (ART) uses a combination of antiretroviral drugs for ongoing suppression of viral replication. Protease inhibitors (PIs) are a crucial component of ART that target the essential HIV protease (PR) enzyme. However, HIV's high replication rate ($10^7$–$10^9$ newly infected cells/day in a patient [1]) facilitates the emergence of drug-resistant strains under the selective pressures of PI therapies that compromise their efficacy. For example, drug resistance to protease inhibitors (PIs) has been shown to emerge in up to 50% of patients [2]. The clinical urgency of this problem is underscored by the fact that even with modern combination ART, treatment failure due to drug resistance remains a significant concern, particularly in resource-limited settings where treatment options may be restricted [3]. Understanding which evolutionary paths are most likely to lead to resistance under different treatment regimens could inform both treatment selection and the design of future antiretroviral drugs. Therefore, it would be helpful to forecast viral evolutionary paths that lead to drug-resistant strains in different ART treatment regimens. However, forecasting protein evolution faces the following challenges.

First, we do not have adequate mechanistic knowledge of the processes that drive evolution. A viral evolutionary path is a sequence of evolving protein genotypes that is driven by evolutionary fitness and the stochastic influences of mutations and genetic drift [4,5]. Existing computational approaches to predicting drug resistance have primarily relied on either genotype-phenotype association models that predict resistance from sequence data [6], or fitness landscape models that attempt to quantify evolutionary constraints [7]. However, genotype-phenotype models typically do not account for the evolutionary pathways by which resistance emerges, while fitness landscape approaches often require extensive experimental data or structural information that may not be available for all drug combinations. Neither approach fully captures how treatment-specific selective pressures shape the accessible mutational space during viral evolution. There is no consensus on the mechanism underlying natural selection because it is defined as a function of *fitness* of different genotypes [8] and there is no broad consensus on the precise biological and

mathematical definition [9]. Previous studies have correlated viral fitness with different properties, for example, protein structural stability, replicative ability, epidemiological fitness, transmissive ability, and enzymatic activity.

Second, the effects of a mutation on the evolution of a genotype depend on the selective pressures in the environment [10] and also on the genotype or the sequence background in which the mutation is introduced [11]. Therefore, even in the same environment or set of selective pressures of a specific ART treatment regimen, a large number of evolutionary paths are possible due to the combined phenomena of stochastic influences and epistasis.

Our contribution in this work is a framework that addresses these challenges in forecasting evolution in different environments of different PI treatment regimens. We briefly outline it as follows. First, we train logistic regression models to learn the coevolutionary information in a set of related protein sequences. Coevolutionary information refers to the signal of correlated mutations between amino acid residues that maintains or refines functionality and/or structure of the protein. It has previously been related to protein structure, function, and fitness [12–15]. We take into account the selective pressures of different PI treatment regimens by training different models for each of the different set of PIs from which observed genotypes were isolated. Consequently, our framework infers probabilities of observing particular residues at a particular position conditioned both on the genotype and the treatment regimen. We use these probabilities to stochastically simulate evolutionary trajectories to subsequently study statistics of emergence of drug-resistant genotypes in different treatment regimens. To this end, we train logistic regression models to infer drug-resistant genotypes using data sets of clinically measured drug resistance. Finally, we study statistics of drug-resistant genotypes in simulated evolutionary trajectories and do a comprehensive ablation analysis to determine which amino acids at which positions are critical for emergence of accessible drug-resistant genotypes during PR evolution in different treatment regimens that conform to the epistatic patterns of the treatment regimen.

## 2 Methods

### 2.1 Genotype-rx data set

Data set of protease isolates from different subjects is taken from https://hivdb.stanford.edu/download/GenoRxDatasets/PR.txt. The treatment given to each subject prior to PR isolation is reported for each isolate as a set of protease inhibitors (PIs), which is a subset of {Atazanavir (ATV), Ritonavir (RTV), Indinavir (IDV), Lopinavir (LPV), Nelfinavir (NFV), Saquinavir (SQV)}. If no PI was given, it is reported as 'None'. We processed the data set by removing PR sequences in three cases. (1) Isolates with insertions, deletions, or gaps in their sequence. (2) Isolates with ambiguity in their sequence—a mixture of amino acids reported at a position in the sequence. (3) Isolates with ambiguity in the treatment received—'Unknown' or 'PI' in the reported treatment. 11 treatment regimens had at least 100 unique PR isolates and are considered in this work (see Table 1 for the list). S1 Table shows the number of filtered unique genotypes extracted from the Stanford database for each treatment regimen.

**2.1.1 Genotype-phenotype drug resistance data set.** We used the genotype-phenotype correlation data set of PR isolates available at https://hivdb.stanford.edu/pages/genopheno.dataset.html. It reports *in vitro* drug-fold resistances [18] of PR sequences to up to 8 different PIs—ATV, DRV, Fosamprenavir (FPV), IDV, LPV, NFV, SQV, and Tipranavir (TPV). We ignored PR isolates with mixtures or X in any position to avoid ambiguous sequences. We annotated the resistance of a PR sequence to a drug as low/intermediate (label 0) or high (label 1) based on the high cut-off thresholds in [19]. We confirmed that there does not exist a PR sequence that was labeled both 0 and 1 with respect to the same drug.

### 2.2 Mathematical model

**2.2.1 Terminology.** Let $\mathcal{F}$ be the collection of treatment regimens in the processed genotype-rx data set and $S_F$ be the set of PR sequences isolated from subjects given the treatment regimen $F \in \mathcal{F}$ with $|S_F| = n_F$,

$$S_F = \{\sigma_1, ..., \sigma_{n_F}\} \tag{1}$$

**Table 1.** Top three positions in PR sequence with the most important positive epistatic interactions in different treatment regimens.

| Treatment regimen | Rank 1 | Rank 2 | Rank 3 |
|---|---|---|---|
| None | 63 | 67 | 20 |
| NFV | 63 | 37 | 36 |
| SQV | 72 | 63 | 69 |
| IDV | 12 | 20 | 63 |
| LPV | 63 | 37 | 12 |
| RTV | 63 | 36 | 69 |
| IDV, NFV | 10 | 82 | 37 |
| ATV, RTV | 63 | 19 | 36 |
| RTV, SQV | 37 | 73 | 20 |
| IDV, RTV, SQV | 20 | 63 | 36 |
| IDV, NFV, RTV, SQV | 82 | 10 | 71 |

where

$$\sigma_i = (\sigma_{i1}, ..., \sigma_{iL}), \quad \sigma_{ij} \in C = \{20 \text{ amino acids}\} \cup \{*\}. \tag{2}$$

$L = 99$ for PR sequences.

Similarly, let $D = \{$ATV, DRV, FPV, IDV, LPV, NFV, SQV, and TPV$\}$ and $S_d$ be the set of PR sequences for which drug-fold resistance to $d \in D$ is reported in the genotype-phenotype data set.

**2.2.2 Training logistic regression models to learn coevolutionary epistatic interactions.** We first compute the onehot encoder for all sequences in $S_F$, and we denote it by $O_F$. The aim is to train a model to infer the probability of observing a specific amino acid at position $i$ conditioned on rest of the sequence. We denote this by $P_F(A_i = \alpha \mid \sim i)$—probability of $\alpha$ at position $i$ conditioned on rest of the sequence or "not $i$" (denoted by $\sim i$). To train a logistic model to infer $P_F(A_i = \alpha \mid \sim i)$, we define inputs as the onehot encoding of all positions but $i$ of sequences in $S_F$ using $O_F$ and the corresponding outputs as the amino acids at position $i$. Python package sklearn [20] was used for both onehot encoding and logistic regression training and inference. Models were trained with $L_2$ regularization that penalizes large parameters to prevent overfitting. Maximum iterations were set to 10000. S2 Table shows the length of the onehot encodings and the number of parameters of epistatic models (cumulative over all positions) for each treatment regimen. S1 Fig shows the distribution of the magnitudes of the parameters of the trained models for each treatment regimen.

**2.2.3 Training logistic regression models for drug resistance inference.** Since resistance to a PI $d \in D$ is inferred for evolutionary trajectories simulated in different treatment regimens $F \in \mathcal{F}$, we cannot onehot encode only the sequences in $S_d$. For every pair of $d$ and $F$, we compute the onehot encoder of all sequences in $S_F \cup S_d$, and we denote it by $O_F^d$. We define inputs to the logistic regression model as the onehot encoding of the sequences in $S_d$ using $O_F^d$ and the corresponding outputs as binary labels for high drug resistance based on thresholds defined in Table 1 in [19].

## 2.3 Simulating evolutionary trajectories

We start with the consensus B PR sequence (https://hivdb.stanford.edu/pages/documentPage/consensus_amino_acid_sequences.html) as the initial sequence. We select a point-mutation stochastically, weighted by the inferred probabilities $(P_F(A_i = \alpha \mid \sim i))^\beta$ for all amino acids $\alpha$ possible at position $i$ and for all positions $1 \leq i \leq L$, where $0 \leq \beta \leq 1$ is a hyperparameter called the inverse temperature. Inverse temperature scaling, also known as importance sampling or tempering, is a well-established computational technique for efficiently exploring rare events in stochastic systems [21,22]. $\beta = 1$ samples directly from the learned probabilities, while $\beta < 1$ flattens the distribution, increasing exploration of low-probability mutations that may be important for adaptation. We apply the point mutation, compute the conditional probabilities for the resulting sequence, and repeat iteratively 1000 times. This gives a single trajectory of protein evolution.

## 2.4 Comparing statistics of drug-resistant genotypes in simulated trajectories

The number of unique drug-resistant genotypes in 1000 trajectories of 1000 steps each is very small at $\beta = 1$, which can skew the statistics of drug-resistant genotypes. A direct approach would be to increase the number and/or length of the trajectories to get more data for statistics of drug-resistant genotypes. However, this approach is computationally inefficient. Instead, we smoothen the statistics of drug-resistant genotypes by simulating trajectories at multiple lower values of $\beta$. Specifically, we simulate 1000 trajectories of 1000 steps for each $\beta \in [0.4, 0.5, 0.6, 0.65, 0.7, 0.75, 0.8, 0.9, 1.0]$ in each treatment regimen.

# 3 Results

## 3.1 Epistatic logistic regression models capture biologically relevant features

### 3.1.1 Simulated sequences maintain proximity to treatment-specific observed sequence space.
We first compare statistical features of the trajectories simulated using the epistatic models with the trajectories simulated using the independent model which learns only the marginal amino acid frequency at each position. Across all 11 treatment regimens, simulated trajectories using the epistatic model maintained significantly closer proximity to observed sequences compared to the independent model—the mean minimum Hamming distance to the nearest observed sequence was 2.3 mutations for the epistatic model versus 5.3 mutations for the independent model. This demonstrates that the epistatic models simulate trajectories that are more biologically plausible in different treatment regimens.

### 3.1.2 Important epistatic interactions correspond to spatially closer residues in the folded protein structure.
We compare magnitudes of coefficients of the trained logistic regression models with spatial distance between the corresponding positions in the folded protein structure. Fig 1A and 1B show results for no treatment and IDV+NFV+RTV+SQV treatment regimen, respectively. We visually observe that positions with high interaction effects are spatially closer. Fig 1C shows the distributions of spatial distances between positions corresponding to top 1% interactions versus the remaining interactions as box plots. Mann-Whitney U test determines that the boxplots are significantly different ($p$-value $\ll 0.001$), and box plots show that the top 1% interactions correspond to positions that are spatially closer in the folded strucutre as compared to the remaining interactions.

### 3.1.3 PageRank centralities of learned epistatic interactions reveal important positions.
We next assess PR positions that are important according to the trained epistatic models. For each treatment regimen $F$, we define two discrete directed graphs—$G_F^+$ for positive interactions and $G_F^-$ for negative interactions. The nodes are the positions of the PR sequence and the edges are weighted as follows. The directed edge from $i$ to $j$ in $G_F^+$ ($G_F^-$) is weighted by the sum of magnitudes of all positive (negative) interactions of position $i$ on $j$. We then rank the nodes or sequence positions by their PageRank centralities in these graphs. Tables 1 and 2 show top three positions in $G_F^+$ and $G_F^-$, respectively, for different treatments regimens. We note the prevalence of importance of position 63 in most cases.

## 3.2 Logistic regression models infer drug resistance with high accuracy

To study the statistics of drug resistant genotypes in simulated PR evolution trajectories, we need mathematical models to infer drug resistance of the genotypes in the trajectories. Hence, we trained logistic regression models using publicly available data sets of clinically measured drug resistance of PR genotypes to 6 different PIs—Atazanavir (ATV), Fosamprenavir (FPV), Indinavir (IDV), Lopinavir (LPV), Nelfinavir (NFV), and Saquinavir (SQV). A separate model is trained for each PI (balanced train:test split of 80 : 20). Table 3 shows that the logistic regression models achieve F1-scores ranging from 0.75 to 0.95 in classifying test genotypes as low or highly drug resistant. Table 4 shows the top ranked coefficients of the trained logistic regression models to predict high drug resistance to different PIs.

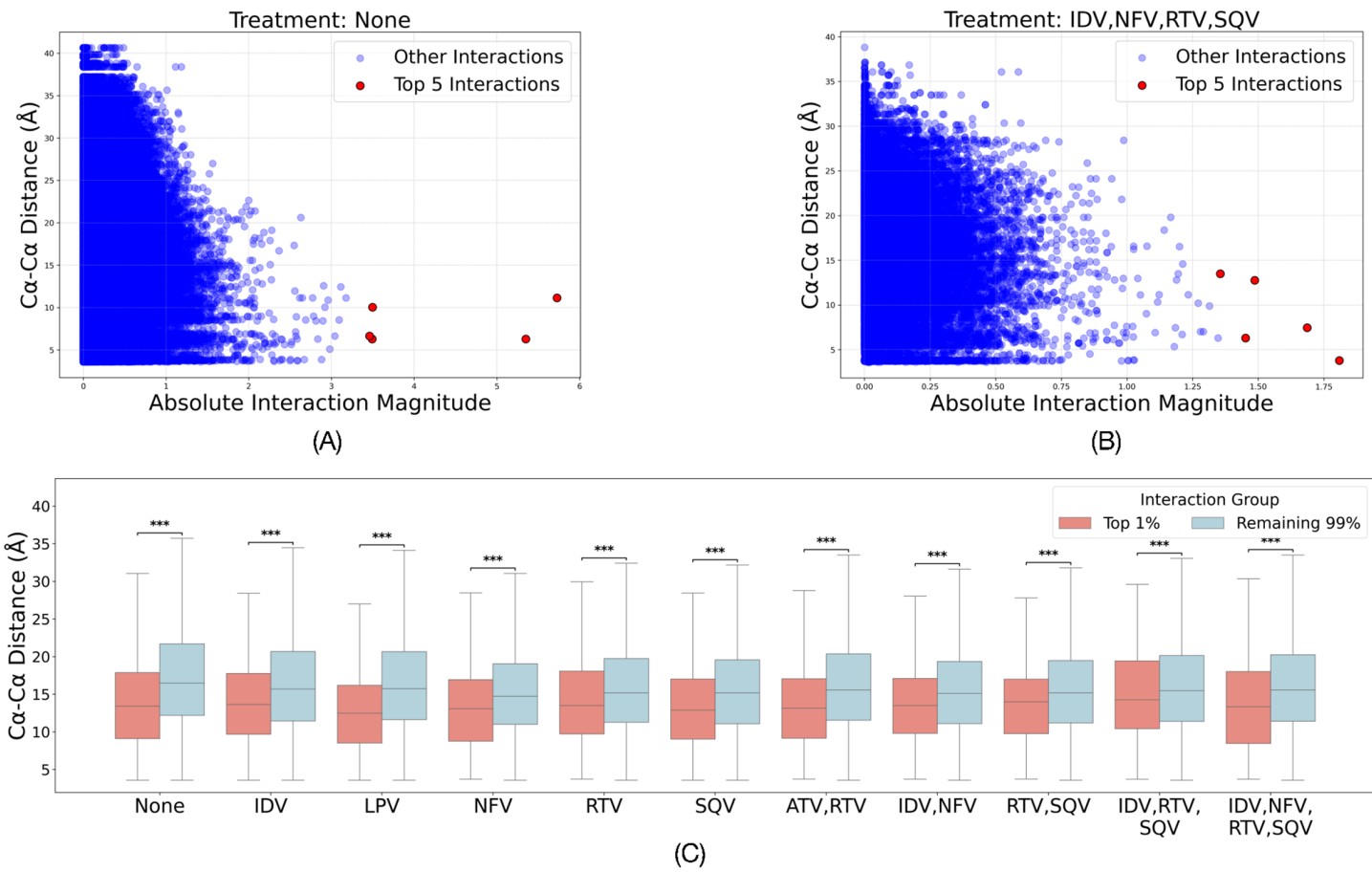

**Fig 1**. **Epistatic interactions preferentially link structurally proximal residues.** (Top) Scatter plots showing the relationship between the magnitude of epistatic interactions and the physical distance between corresponding residue pairs in the folded structure. Plots are shown for (A) no treatment and (B) IDV+NFV+RTV+SQV treatment regimen. Top five interactions are predominantly found between residues that are spatially close. (C) Distribution of spatial distances corresponding to top 1% interactions are significantly different from the remaining interactions (Mann-Whitney U test, \*\*\* indicates *p*-value < 0.001), with the former having lower values, validating that the epistatic models are learning likely functionally relevant constraints.

**Table 2**. **Top three positions in PR sequence with the most important negative epistatic interactions in different treatment regimens.**

| Treatment regimen | Rank 1 | Rank 2 | Rank 3 |
|---|---|---|---|
| None | 61 | 12 | 69 |
| NFV | 63 | 37 | 69 |
| SQV | 63 | 61 | 37 |
| IDV | 89 | 63 | 37 |
| LPV | 63 | 19 | 12 |
| RTV | 12 | 69 | 82 |
| IDV, NFV | 37 | 10 | 72 |
| ATV, RTV | 63 | 82 | 12 |
| RTV, SQV | 36 | 37 | 73 |
| IDV, RTV, SQV | 63 | 82 | 72 |
| IDV, NFV, RTV, SQV | 37 | 20 | 71 |

**Table 3**. **F1-scores for drug resistance inference in test data: Number of unique genotypes in the classes of high and not-high drug resistance to 8 different drugs.** Logistic regression models were trained to infer drug resistance to drugs that had at least 100 samples in both classes of drug-resistance. F1-scores of the trained models on test data are shown.

| Drug | FPV | ATV | IDV | LPV | NFV | SQV | TPV | DRV |
|---|---|---|---|---|---|---|---|---|
| Low/intermediate drug res seqs | 679 | 397 | 632 | 576 | 465 | 647 | 368 | 330 |
| High drug res seqs | 153 | 163 | 230 | 132 | 418 | 222 | 37 | 19 |
| F1 score on test data | 0.75 | 0.86 | 0.77 | 0.73 | 0.95 | 0.85 | | |

**Table 4**. **Top 5 coefficients of logisitic regression models trained to predict high resistance to different PIs.**

| ATV | DRV | FPV | IDV | LPV | NFV | SQV | TPV |
|---|---|---|---|---|---|---|---|
| 50L | 32I | 50V | 84A | 10F | 88S | 84A | 47V |
| 84A | 33F | 84A | 50I | 84A | 30N | 48V | 82L |
| 48V | 50V | 84C | 48V | 50V | 84C | 84C | 84V |
| 20T | 89V | 16A | 82F | 20T | 90M | 90M | 10V |
| 35N | 10F | 33F | 95F | 47A | 84A | 53L | 35N |

### 3.3 Analyzing mutation propensity and reachability of drug resistant genotypes in simulated trajectories

We study two statistics of the drug resistant genotypes as follows. First, given a sequence $\sigma = (\sigma_1, ..., \sigma_L)$, we say that it has a high propensity to mutate in treatment regimen $F$ if $\min_{1 \leq i \leq L} P_F(A_i = \sigma_i | \sim i)$ is small. In other words, if the least probable $(i, \sigma_i)$ pair has low probability, then the genotype has high propensity to mutate. For notational convenience, we define mutation propensity $\sigma$ in treatment regimen F as $m_F(\sigma) = -\min_{1 \leq i \leq L}\{\log P_F(A_i = \sigma_i | \sim i)\}$, such that $\sigma$ has a lower mutation propensity than $\omega$ if $m_F(\sigma) < m_F(\omega)$. Fig 2A shows that PR genotypes isolated from multi-drug PI treatment regimens have lower mutation propensity compared to those from mono-therapy or drug-naive contexts. On the other hand, Fig 2B shows that drug-naive genotypes have increased mutation propensity in multi-drug PI environments compared to the drug-naive baseline, with the highest increase observed for the four-drug combination (IDV+NFV+RTV+SQV). This suggests that drug-naive sequences are poorly adapted to multi-drug selective pressures.

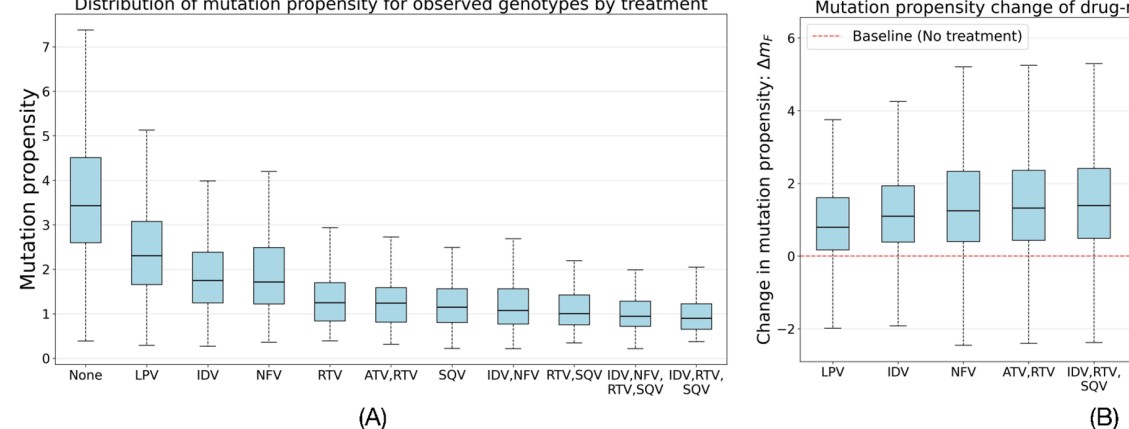
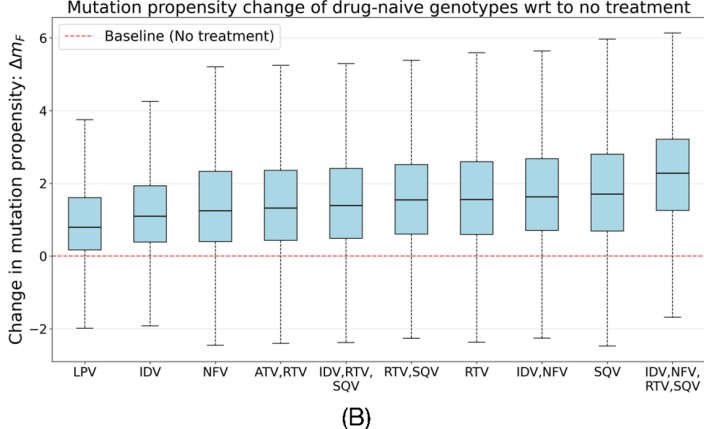

**Fig 2**. **Mutation propensity distinguishes adapted from unadapted genotypes.** (A) Mutation propensities of observed genotypes evaluated within the treatment regimen from which they were isolated. Genotypes from multi-drug PI regimens exhibit significantly lower mutation propensity than those from mono-therapy or drug-naive contexts. (B) Change in mutation propensity of drug-naive genotypes when evaluated under various PI treatment regimens relative to the no-treatment baseline. Naive genotypes show a marked increase in mutation propensity, signifying they are poorly adapted to the selective pressures imposed by PIs.

The second property that we consider is the probability of reaching a genotype along evolutionary trajectories. This depends on the probabilities of all mutational events along the trajectory up to the first occurrence of the genotype. We define 1/reachability of a genotype $\sigma$ in an evolutionary trajectory $T$ as $q(\sigma, T) = \log(-\sum_{1 \leq i \leq t} \log P_i)$ where $P_i$ is the inferred probability of the mutation event that was selected at step $i$, and $t$ is the first occurrence of $\sigma$ in $T$. We define $q$ as 1/reachability because higher $q$ implies lower reachability.

Fig 3A shows the distribution of mutation propensities and 1/reachability of genotypes resistant to NFV in evolutionary trajectories simulated in the treatment regimen IDV+RTV+SQV. Arguably, drug resistant sequences with low mutation propensity and high reachability are of biological interest. We combine both these criteria into a scalar using Pareto optimality as follows. We rescale both criteria to [0,1] by dividing mutation propensity by 20 and 1/reachability by 10. We define Pareto optimality, $\phi(\tau)$, as the number of genotypes in the distribution with the $L_2$ norm of the rescaled mutation propensity and 1/reachability at most $\tau$. Fig 3B illustrates the contour of $\tau = 0.45$ and genotypes that would be counted in $\phi(0.45)$ (marked by red). Fig 3C shows the plot of $\phi(\tau)$ as a function of $\tau$. We show next how we use this plot as a quantitative measure to compare the landscape of accessible resistant genotypes across different treatment regimens.

Fig 4A shows that $\phi(\tau)$ in the treatment regimen IDV+RTV+SQV is higher than the no treatment regimen for all values of $\tau$. We define fractional change in $\phi(\tau)$ with respect to a reference $\phi^*(\tau)$ as $\frac{\phi(\tau)+\epsilon}{\phi^*(\tau)+\epsilon}$. $\epsilon$ stabilizes the fraction at small counts, and we picked $\epsilon = 10$ since it is much smaller than $1e6$, the scale of cumulative counts of drug-resistant genotypes. Fig 4B shows that the fractional increase with respect to no treatment is not a uniform function of $\tau$ and is the most significant for $0.2 < \tau < 0.6$. Fig 4C shows fractional changes for all 11 treatment regimens with respect to no treatment, labeled based on mono-PI and multi-PI treatment regimens. Multi-PI treatments regimens show higher fractional change as compared to mono-PI treatment regimens, with the exception of ATV+RTV treatment.

### 3.4 Mutations critical for resistance to different PIs in different treatment regimens are revealed

We determine the importance of point-mutations in the emergence of drug resistance by analyzing the fractional change in $\phi(\tau)$ when a specific amino acid residue $\alpha$ is not allowed at a specific position $i$ is not allowed with respect to when all

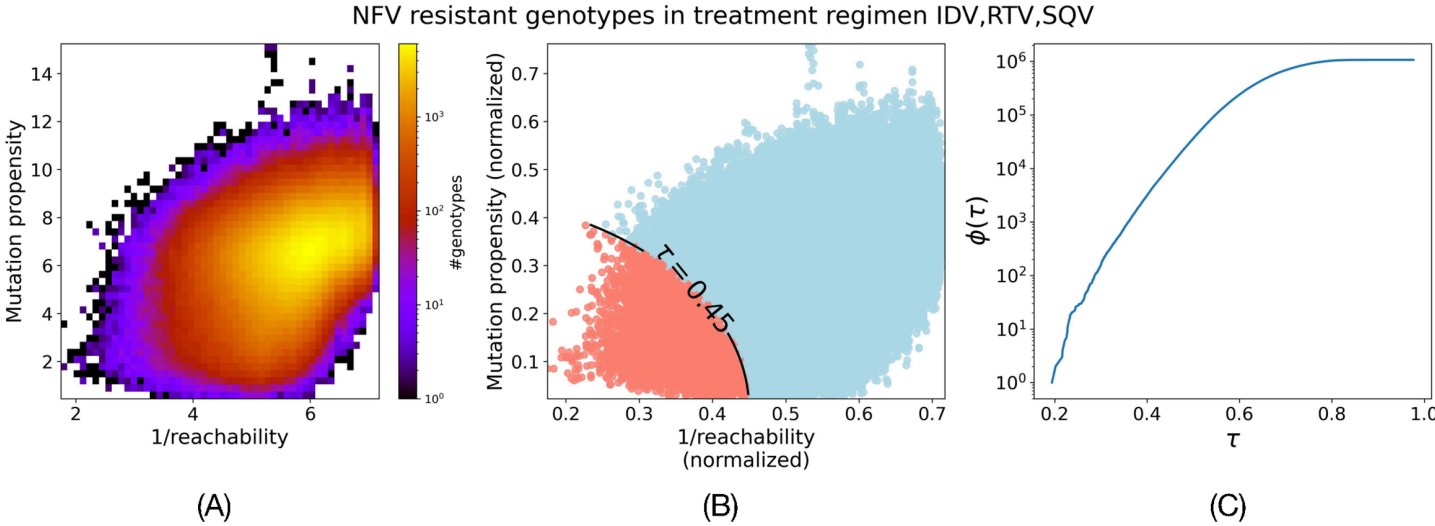

(A)   (B)   (C)

**Fig 3. Analyzing statistics of mutation propensity and 1/reachability simultaneously.** (A) 2D heatmap shows the joint distribution of mutation propensity and 1/reachability for NFV resistant genotypes under the IDV+RTV+SQV treatment regimen. Drug resistant genotypes of high biological interest are those with low mutation propensity and high reachability. (B) Example of contour of $\tau = 0.45$. $\phi(\tau)$ at $\tau = 0.45$ is the cumulative count of the genotypes with norm of their mutation propensity and 1/reachability less than 0.45. These genotypes are shown as red points. (C) Plot of $\phi(\tau)$ quantifies how the number of genotypes increases as $\tau$ increases.

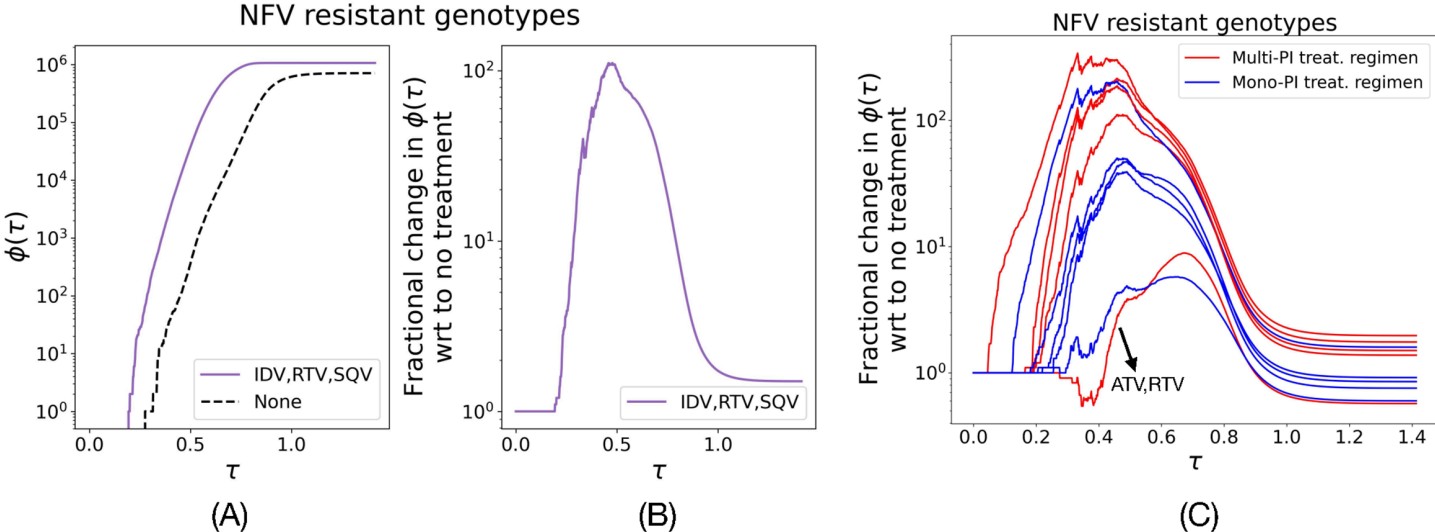

**Fig 4**. **Number of resistant genotypes with low mutation propensity and high reachability is higher by several orders of magnitude in most multi-PI treatment regimens.** (A) $\phi(\tau)$ is higher for IDV+RTV+SQV treatment regimen as compared to no treatment. (B) Fractional change in $\phi(\tau)$ with respect to no treatment as a function of $\tau$ shows an increase by multiple orders of magnitudes under IDV+RTV+SQV treatment. (C) Fractional changes of all treatment regimens with respect to no treatment. Multi-PI treatment regimens are marked red and mono-PI treatment regimens are marked blue. Multi-PI treatment regimens have a higher number of accessible resistant genotypes by several orders of magnitude, especially in the range of $0.2 < \tau < 0.6$. ATV+RTV (annotated) is the outlying multi-PI treatment with relatively low fractional increase.

mutations are allowed. We call this leaving out $\alpha$ at $i$ or leave out $i\alpha$. We do this for all possible combinations of residues and positions. Fig 5 shows fractional changes in $\phi(\tau)$ for all leave-out cases for different treatment regimens and drug-resistant genotypes. We show results for all multi-PI treatment regimens except for ATV+RTV since it showed relatively low fractional change in $\phi(\tau)$ with respect to no treatment (Fig 4C). We identify and label top three outlier curves in the figures such that leaving out the corresponding $i\alpha$ significantly reduced the number of drug-resistant genotypes (fractional change $\ll 1$) for $\tau < 0.6$ (low mutation propensity and high reachability). We find that the outliers are for leaving out 90M, 10I, 30N, 63P, 54V, 71V, 84V, 84A, and 84C. With respect to the consensus B type, these correspond to the point-mutations L90M, L10I, D30N, L63P, I54V, A71V, and I84V/A/C, respectively. Importantly, the analysis revealed drug-specific dependencies. For example, the prohibition of P at 63 and of N at 30 show a disproportionately large effect on the development of resistance to NFV compared to other protease inhibitors.

## 4 Discussion

We used logistic regression models to learn epistatic interactions and infer transition probabilities in a given sequence background and under different PI treatment regimens. We introduced mutation propensity of a genotype that is a model-specific metric to quantify conformity of the genotype to the treatment regimen. We trained another set of logistic regression models to infer resistance of PR genotypes to different PIs. We combined simulated evolutionary paths with drug-resistance inference to reveal that reachability of drug-resistant genotypes with low mutation propensity increase with the number of PIs in the treatment regimen, with the exception of ATV+RTV. Finally, we did a comprehensive leave-one-out analysis and determined that allowing 90M, 10I, 30N, 63P, 54V, 71V, 84V, 84A, and 84C is critical to higher reachability of drug-resistant genotypes with low mutation propensity. With respect to the consensus B type, these correspond to the mutations L90M, L10I, D30N, L63P, I54V, A71V, and I84V/A/C, respectively.

Our model identifies a cohort of mutations whose importance in drug resistance is well-corroborated by biological evidence. Mutations D30N, L90M, and I84V/A/C are highlighted as critical. These are canonical primary resistance

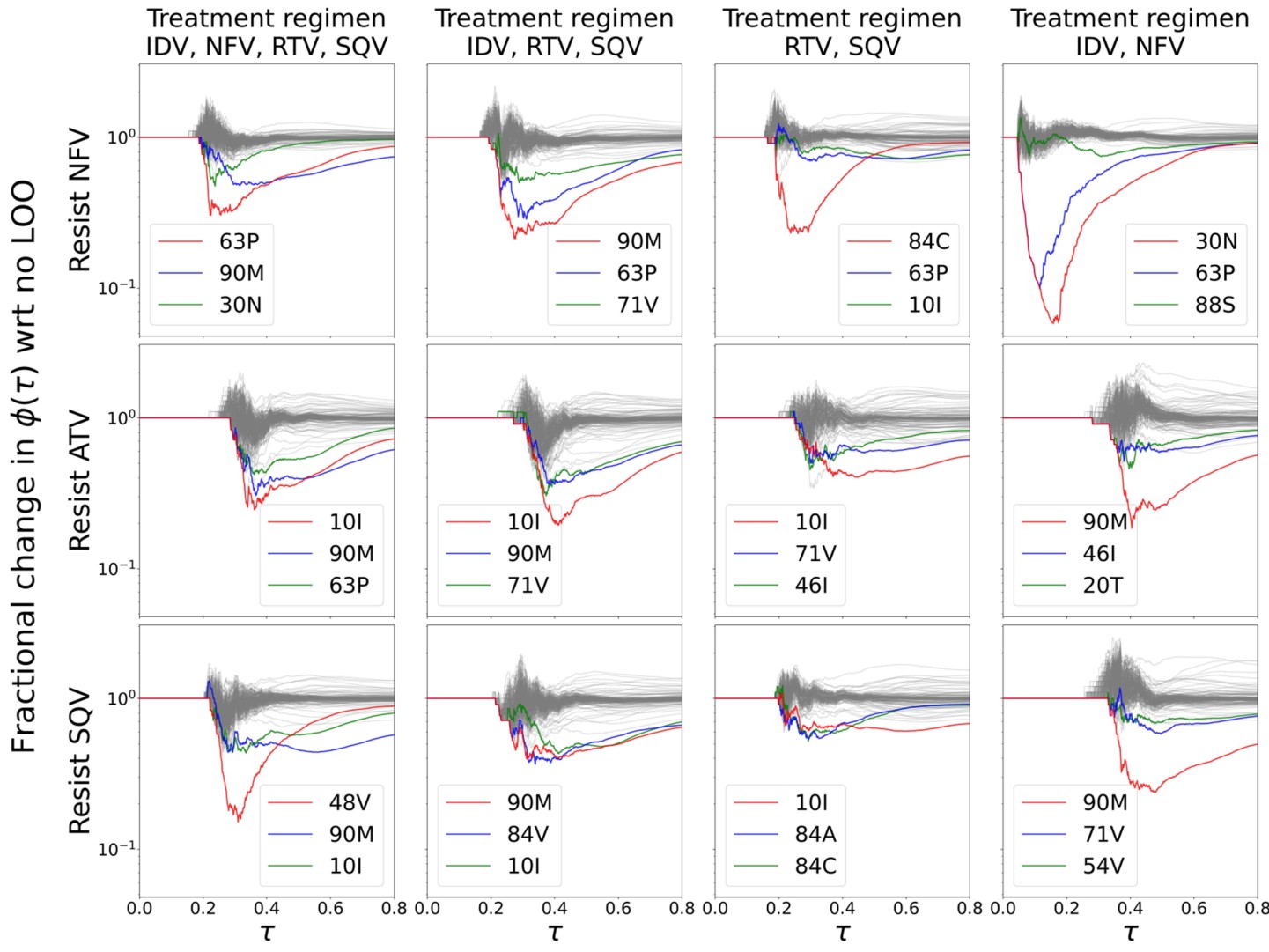

**Fig 5**. **Leave-one-out (LOO) analysis identifies point mutations critical for the emergence of accessible drug-resistant genotypes in different treatment regimens.** $\tau$ is the $L_2$ norm of the rescaled mutation propensity and 1/reachability and $\phi(\tau)$ is the number of drug-resistant genotypes with $L_2$ norms of their mutation propensity and 1/reachability at most $\tau$. Each panel shows the fractional change in $\phi(\tau)$ as a function of $\tau$ when a single point mutation is prohibited from occurring during the simulations, relative to the control where all mutations are allowed (no LOO). A drop in the curve (to values $\ll 1.0$) indicates that the prohibited mutation is critical for accessing the resistant phenotype. Top three ablations that resulted in the most significant drops are highlighted with different colors and shown in the legends.

mutations that arise within the protease active site, directly impairing inhibitor binding [23,24]. The I84V mutation, in particular, is known to confer broad cross-resistance to multiple PIs, and its identification by the model underscores the selection pressure exerted by combination therapies [25]. Concurrently, the model pinpoints I54V, L10I, and A71V, which are recognized as secondary mutations that compensate for the fitness cost of primary mutations, thereby modulating the overall resistance phenotype [26].

Another validation of our model is its ability to move beyond identifying a general suite of resistance mutations to elucidating specific, context-dependent relationships between drugs and individual mutations. This is most evident in the case of NFV. Our leave-one-out analysis (Fig 5) shows that prohibiting the primary mutation D30N virtually eliminates the evolutionary accessibility of NFV-resistant genotypes, confirming its role as the signature mutation for this drug [27]. Crucially,

the model also identifies the polymorphic mutation L63P as highly critical specifically for the NFV resistance pathway, a dependency not observed for other PIs like ATV. This result precisely recapitulates the well-documented D30N-L63P co-evolution, where L63P acts as a compensatory mutation that restores the fitness costs incurred by D30N, thereby making the primary resistance pathway viable [28]. Interestingly, 63P is not among the top five coefficients in the logistic regression model that predict high resistance to NFV (see Table 4), but it is among the top 3 positions in positive and/or negative epistatic interactions in most treatment regimens (Tables 1 and 2. This shows the importance of our framework in studying the complex interplay between both evolution and emergence of drug resistance.

We discuss the outlying behavior of the dual therapy of ATV+RTV as compared to other multi-PI treatment regimens. There is very little fractional increase in drug resistant genotypes in this treatment regimen. Fig 6 shows results of LOO

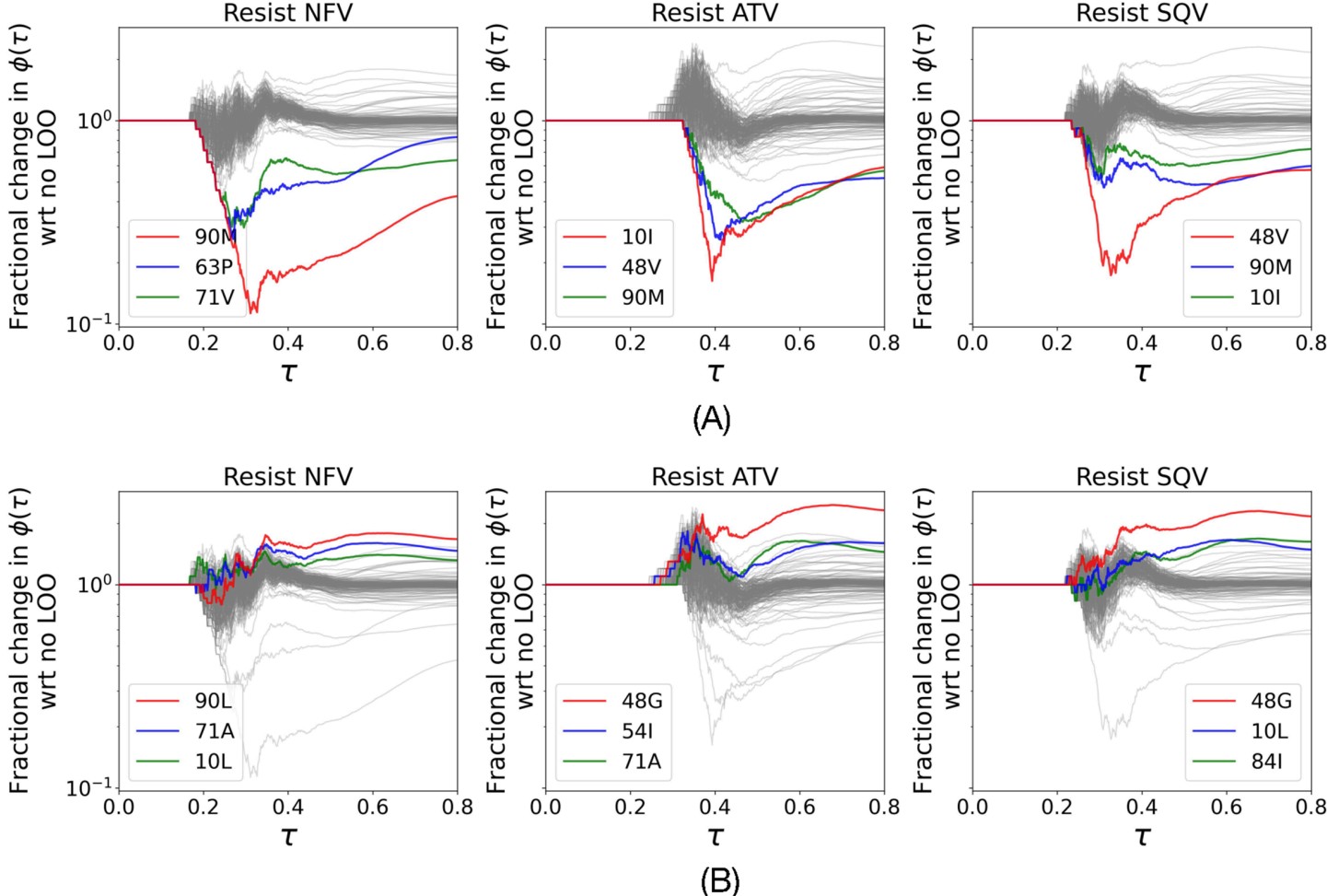

**Fig 6**. **Analysis of resistance-promoting and -suppressing mutations under ATV+RTV therapy.** $\tau$ is the $L_2$ norm of the rescaled mutation propensity and 1/reachability and $\phi(\tau)$ is the number of drug-resistant genotypes with $L_2$ norms of their mutation propensity and 1/reachability at most $\tau$. Each panel shows the fractional change in $\phi(\tau)$ as a function of $\tau$ when a single point mutation is prohibited from occurring during the simulations, relative to the control where all mutations are allowed (no LOO). (A) Critical resistance mutations whose prohibition decreases the number of accessible resistant genotypes (fractional change <1), indicating they are essential for the resistance pathway. (B) Resistance-suppressing mutations whose prohibition increases the number of accessible resistant genotypes (fractional change >1), indicating their presence is protective against drug-resistance. Top three ablations that resulted in the most significant (A) drops and (B) gains are highlighted with different colors and shown in the legends. Note that this analysis reveals that consensus 71A is resistance suppressing and the mutant 71V is resistance inducing.

analysis of this treatment regimen. Top panel of the figures highlights mutations that are important for drug resistant genotypes and the bottom panel highlights mutations that are important to not get drug resistant genotypes (fractional change is significantly more than 1 when we leave these out). We note that leaving out 71V decreases and leaving out 71A increases emergence of drug resistance to NFV. This is interesting because A at position 71 is the consensus and A71V is a known drug resistant mutation, and hence, the consensus is important to suppress the emergence of NFV-resistant genotypes. We observe similar examples of L10I and G48V for emergence of SQV-resistant genotypes under ATV+RTV treatment.

The results of critical mutations are based on simulated evolutionary trajectories that start at the PR consensus subtype B sequence. We cannot claim that these results generalize for different subtype genotypes as the starting sequence without further investigation. Repeating the leave-one-out analysis with different starting subtype sequences is computationally expensive. A direction of future research is to repeat the analysis for generalization of results in this work to different subtypes and/or to determine computationally feasible approaches to do the same.

The publicly available data sets used in this work are limited to treatment regimens with single or multiple drugs from one of four drug classes—protease inhibitors (PIs), non-nucleoside reverse transcriptase inhibitors (NNRTIs), nucleoside reverse transcriptase inhibitors (NRTIs), and integrase strand transfer inhibitors (INIs). However, regimens consisting of three drugs of different classes, called HAART (Highly Active Antiretroviral Therapy), have been found to be more effective [29–31], with one of the advantages being a reduction in the emergence of drug resistance. Our approach could be applied to data from HAART treatment regimens to determine the mechanisms that alleviated drug resistance in this combination therapy, insights that may be useful in improving treatment regimens.

Recently, treatment regimens consisting of two drugs have been recommended due to their reduced adverse effects and toxicities compared to three-drug regimens [32]. Our analysis predicts that the ATV+RTV treatment regimen has low chances of emergence of drug resistance. However, as with all data-driven inference in complex biological processes, only clinical data can confirm the efficacy of this regimen compared with others.

## Supporting information

**S1 Table. Numbers of filtered unique genotypes extracted from the Stanford HIV database for each treatment regimen.**
(TIFF)

**S2 Table. Lengths of onehot encoded vectors and numbers of parameters of epistatic models (cumulative over all positions) for all treatment regimens.**
(TIFF)

**S1 Fig. Distributions of magnitudes of parameters (cumulative over all positions) of trained models for all treatment regimens.**
(TIFF)

## Acknowledgments

We thank Dr. Wolfgang Resch for helping us in utilizing the computational resources of the NIH HPC Biowulf cluster (http://hpc.nih.gov).

## Author contributions

**Conceptualization:** Manu Aggarwal, Vipul Periwal.

**Formal analysis:** Manu Aggarwal.

**Funding acquisition:** Vipul Periwal.

**Investigation:** Manu Aggarwal.

**Methodology:** Manu Aggarwal, Vipul Periwal.

**Project administration:** Vipul Periwal.

**Resources:** Vipul Periwal.

**Software:** Manu Aggarwal.

**Supervision:** Vipul Periwal.

**Validation:** Manu Aggarwal.

**Visualization:** Manu Aggarwal.

**Writing – original draft:** Manu Aggarwal.

**Writing – review & editing:** Manu Aggarwal, Vipul Periwal.

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
