## [Decision Letter · Decision Letter 0]

25 Jun 2025

PCOMPBIOL-D-25-00787

FORECASTING DRUG RESISTANT HIV PROTEASE EVOLUTION

PLOS Computational Biology

Dear Dr. Aggarwal,

Thank you for submitting your manuscript to PLOS Computational Biology. After careful consideration, we feel that it has merit but does not fully meet PLOS Computational Biology's publication criteria as it currently stands. Therefore, we invite you to submit a revised version of the manuscript that addresses the points raised during the review process.

Please submit your revised manuscript within 60 days Aug 25 2025 11:59PM. If you will need more time than this to complete your revisions, please reply to this message or contact the journal office at ploscompbiol@plos.org. Please include the following items when submitting your revised manuscript:

We look forward to receiving your revised manuscript.

Kind regards,

Jessica M. Conway

Academic Editor

PLOS Computational Biology

Marc Birtwistle

Section Editor

PLOS Computational Biology

**Journal Requirements:**

At this stage, the following Authors/Authors require contributions: Manu Aggarwal, and Vipul Periwal. Please ensure that the full contributions of each author are acknowledged in the "Add/Edit/Remove Authors" section of our submission form.

5) We have noticed that you have uploaded Supporting Information files, but you have not included a list of legends. Please add a full list of legends for your Supporting Information files after the references list.

6) Please note that your Data Availability Statement is currently missing the repository name, and the DOI/accession number of each dataset OR a direct link to access each dataset. If your manuscript is accepted for publication, you will be asked to provide these details on a very short timeline. We therefore suggest that you provide this information now, though we will not hold up the peer review process if you are unable.

7) Please amend your detailed Financial Disclosure statement. This is published with the article. It must therefore be completed in full sentences and contain the exact wording you wish to be published.

8) Please send a completed 'Competing Interests' statement, including any COIs declared by your co-authors. If you have no competing interests to declare, please state "The authors have declared that no competing interests exist". Otherwise please declare all competing interests beginning with the statement "I have read the journal's policy and the authors of this manuscript have the following competing interests"

**Reviewers' comments:**

Reviewer's Responses to Questions

**Comments to the Authors:**

Reviewer #1: The paper considers profiles of mutations in the protease enzyme in drug-naive and drug-treated HIV samples. It builds a probabilistic profile of the different amino acids that appear at all loci throughout the protein in different data subsets, and uses this profile to build a model for the "mutation propensity" of a given genotype under a given drug treatment. This mutation propensity is both interrogated itself, intepreted as linked to fitness, and also used to simulate putative evolutionary trajectories under different circumstances. A combination of these pictures is used to build a picture of genotypes that are both evolutionarily accessible and high fitness (under the promises of the model). Finally, a collection of particular mutations that are predicted to challenge the emergence of accessible, fit genotypes are identified.

This is an interesting paper and the appearance of previously-identified mutations of importance in the final analysis provide some support for its validity. There's also an interesting identification of a known, but less understood, mutation as a particularly important "driver" of drug resistance evolution. There is interesting science here. But it is, at the moment, obfuscated by a combination of unintuitive writing, confusing methods, and a collection of results that distract from (rather than support) the main scientific picture. Fixing these would both focus on the key ideas and make them more interpretable.

A collection of comments are below, starting with some cross-cutting general ones and then some more specific points. For transparency, I am Iain Johnston and I am happy for this review to be treated as public domain. To my eyes, my most important shortcomings as a reviewer here are a lack of familiarity with the specific "FEM" inference method central to the paper, and a lack of experience with the molecular biology of HIV protease.

Top-level comments

1. The most important variables throughout the manuscript are called things like

"negative log likelihood to not mutate" / "FEM-fitness"

"log(-log prob to reach)"

and the collection of double negatives involved really challenges intuitive intepretation. I strongly suggest the authors introduce shorthand for these quantities (perhaps things like "mutation propensity" and "accessibility" respectively) to help at-a-glance interpretation of the text and figures. I particularly think "FEM fitness" should be avoided, as it pictures fitness as forward-looking (propensity for future mutation). I don't think any of the scientific story suffers if "[predicted] mutation propensity" is used instead.

2. Section 2.2, and other parts to a lesser extent, present results which are largely or wholly dependent on a model variable (beta) which I find very hard to tie to a biological observable. It's not clear how much this adds to the scientific story, and in later sections it's not clear what particular choice of this value has been used in evolutionary simulation. Likewise, the large set of PCA visualisations in Fig 1 don't provide much (any?) scientific insight and can readily be removed.

3. The core technology is free energy minimisation inference, but the positioning of the introduction and the rest of the manuscript with respect to this is very confusing. The technology looks somewhat like a modification of logistic regression, particular as there is no scientifically meaningful "energy". The training approach is never described, and the key quantity P_F is conditioned on a particular point in a potentially vast state space that cannot possibly be spanned by the training set. I may well be misunderstanding, but I can't see how the steps in Eqns. 17-19, which seem very important for the core quantity of interest, follow (see specific comment below). More information about these points is needed -- see individual points below.

4. Throughout, the reader is led to wonder if a simpler set of approaches would give comparable results. What if we used logistic regression instead of FEM inference? What if the classification model in Sec 2.3 just contained the core set of amino acid mutations found later as predictors (this analysis should be done)? What if "evolutionary accessibility" was just summarised as number of mutations from the reference case? And importantly -- how important actually is epistasis in this system? We are given (unless I'm misunderstanding) no information about how important this conditioning on current genotype actually is for the outputs of the model?

More targetted comments

Line numbers would have helped this review!

Second para p1 "First". A couple of strange directions. Why is selection deterministic when mutation and drift are stochastic? Fundamentally it is applied by a noisy environment. As a description of evolutionary models this holds; but we are not yet talking about maths, just noisy biology! And -- fitness is generally defined as replicative ability. The determinants of *that* are hugely complicated.

p2

"the effects of a mutation on the properties of a genotype depend on the selective pressures in the environment". This is either true or false depending on what "properties" means, and most reasonable meanings would make it false. The effect of mutation on a genotype don't depend on selection; the effect of mutation on a genotype's later representation in a population do, but via phenotype.

"a specific point-mutation can theoretically have up to 20^L different effects on protein genotypes of length L since there are 20 different amino acids." Not if we're talking about SNPs (most common definition of point mutation), because only a small subset of those 20 amino acids will be accessible via a single polymorphism. Should clarify that you mean point mutation at the protein sequence level.

"mutations are randomly selected but weighted by inferred transition probabilities." -- this is very unclear to a reader who is at this point. What does randomly selected mean? What does selected mean if the approach "avoids a mechanistic formulation of natural selection"? What does weighted mean?

"we do not use an energy model" -- given that you are using free energy minimisation, this is also very confusing at this point and needs more explanation.

"forecast the emergence of drug-resistant PR strains that have high FEM-fitness, defined specifically and precisely for our approach in the following" -- but you've previously said that you will avoid a mechanistic formulation of fitness and an energy model. At this point I have no idea what shape the approach is going to have!

"We define FEM-fitness of a genotype as its likelihood to mutate" -- again confusing, especially as evolution famously cannot look ahead, so fitness cannot possibly be defined by something that will happen in the future. Even if it could, it's not clear what direction this picture works in. Is mutating good (to evolve resistance) or bad (because the vast majority of mutations will be detrimental)?

The use of "log likelihood" is quite confusing here (and throughout). Unless I am misunderstanding, P_F is a probability produced by a trained model, not a likelihood (which is usually used to refer to a probabilistic connection to a particular empirical observation). Why is it a log likelihood and not a log probability? And in fact, as it seems lots of followup will be concerned with comparing drug-treated and drug-naive conditions, why not use a log odds?

p3

Effectively all annotation in Fig 1 is too small to read on a reasonable scale. The plotting protocols should be included in the caption (e.g. that 1A-D use PCA).

I can't follow the "negative loglik of genotypes to not mutate" -- there are too many negatives here. The definition of the previous page refers to the relative probabilities of mutation under two different treatment regimes -- it doesn't refer to probabilities of not mutating.

I half assume that what we are supposed to take from Fig 1 is that the density of points in 1E is above the diagonal, meaning that mutation propensity is systematically higher under drug treatments (vertical axis) than under no treatment (horizontal axis)? I cannot see that 1A-D tell us anything about the science? Any structure in PCA space is not mentioned as a scientific message. Unless I am mistaken, they could readily be removed to focus on the important message 1E.

"Briefly" -- we need more information here! Put this in the methods and expand upon it. Especially how beta informs the "importance sampling" -- it's not at all clear that this in fact importance sampling as opposed to a simple weighted sampling of discrete events. What I think is happening is that you are randomly choosing loci to mutate according to some function of their predicted mutation propensity.

p4

Fig 2 and Sec 2.2 focus on how the evolutionary simulation depends on beta. But it's not at all clear how this provides a link to the underlying science. What is the real-world observable that beta corresponds to? It seems to me to provide a weighting between a null model where every mutation is equally likely (high temperature) and a more constrained model where mutation probabilities are entirely determined by your fitted model (low temperature). But nature doesn't know about your fitted model, so how should I intepret beta when thinking about the biology?

The section label for 2.3 is misleading -- it's for one particular drug resistance, and the score is rather low for all others. Better to remove the specific value (or at least say the range).

p5

Fig 3B is redundant with and less informative than 3C; remove it. But why is the collection of points at exactly the same y-value (around 14) in 3B not reflected in 3C, and what is its meaning?

What is beta in these simulations?

For an informative message about the influence of drug treatment here, it would be useful to have plots like 3A and 3C but under the no-treatment regimen, so that we can see the probabilities of NFV resistance emerging without the drug treatment.

p6

The Pareto optimality section directs us to the Methods to learn about what's going on here, but I can't see any information there about it?

I am completely stuck with:

Briefly, we say that a genotype has Pareto optimality ϕ if the L2 norm of the log likelihood of not mutating and the log log probability of reaching is ϕ. The features are rescaled to [0, 1] by dividing them by respective upper bounds of 20 and 10. Hence ϕ ∈ [0, 2].

What I think this is trying to do is combine the "reachability" of a genotype -- I think under the NFV drug treatment, though this isn't clear -- with its probability of being drug resistant.

Without fully understanding this I'm not sure about Fig 5. I think what we're seeing is that treating with lots of drugs at once gives more genotypes that combine this reachability and resistance probability.

p7

It is interesting that this picture pulls out a subset of mutations that are critical for reachability-resistance. If you now take presence/absence of individuals in this set as predictors in another classification model for resistance, does it do better or worse than your approach in Sec 2.3?

Methods

--------------

In Eqn. 3 it would help to explicitly define u_j as the cardinality of the alphabet of symbols found at position j (assuming I'm right in this intepretation?).

In Eqn. 8 should a_N1 be a_L1? If not then I am misunderstanding the indexing here and some more explanation would be useful.

In Eqn. 11 should e^omega H on the numerator be e^H? Otherwise it doesn't match Eqn. 21.

FEM via Eqn. 11 looks very similar to logistic regression, if sigma_j are considered as binary predictor variables and b an intercept? Can the authors comment on why logistic regression isn't an appropriate formalism for this picture? Also -- free energy minimisation is a much more general process than this specific inference method. Even if that's the name the original authors proposed, it would help to disambiguate this.

I'm worried that Eqns. 17-19 contain some fundamental issues (but I may well be misinterpreting). In going from Eqn. 17 to Eqn. 18 we seem to be saying that a vector can be written as the intersection of a collection of sets, each of which contains a single element of the vector. e.g. (1, 0, 0, 1) = {1} \cap {0} \cap {0} \cap {1}. This can't possibly be what this means but I can't see how else this is meant to be interpreted.

Then in going from Eqn. 18 to Eqn. 19 we seem to be saying that the (joint) probability of a collection of things (whatever the things are, from my previous question) can be written as the product of probabilities of the individual things. But this is only true if the things are independent, and the elements of a onehot vector are certainly not independent -- fixing a 1 means that all the other elements are immediately known. What am I misunderstanding here?

The most important issue is a (complete) omission of the training process. Without this I simply cannot understand how we can learn from this picture. There is a vast space of \tilde{sigma}_\tilde{p} (I'll write this as s_p to avoid pseudo-TeX) vectors, relatively limited training data (a few hundred in some cases, as mentioned in the intro), and it seems very possible that every datapoint might have a unique s_p. Given that the fundamental quantity of interest P_F is conditioned on a particular s_p, it seems very unclear that a collection of training s_p can inform a (disjoint) set of test s_p?

Reviewer #2: Reviewer Comments

Title: FORECASTING DRUG RESISTANT HIV PROTEASE EVOLUTION

Manuscript id: PCOMPBIOL-D-25-00787

Overall Recommendation: Major Revision

This manuscript presents a probabilistic large-deviation model leveraging Free Energy Minimization (FEM) to simulate HIV protease evolution under selective drug pressure and forecast resistance emergence. While the approach is conceptually innovative and addresses an important biomedical challenge, the manuscript requires major revision due to insufficient methodological clarity, validation gaps, and limited biological interpretation in key sections.

Major Comments:

1. FEM-Fitness Metric Requires Clearer Justification Across Regimens

The definition of FEM-fitness (Section 2, page 2) as the minimum negative log conditional probability is central to many downstream claims. However, it remains unclear how this metric accounts for differing selective pressures and sampling biases across treatment regimens. Without normalization or cross-regimen calibration, comparisons in Figures 1D–E and Figure 5 may be confounded.

2. Lack of Baseline Comparisons for Resistance Prediction Models

The FEM-based resistance classifiers (Table 1) report high F1 scores for some drugs. However, the manuscript lacks comparison between established ML models such as random forests, SVMs, or logistic regression. Without benchmarking, it is difficult to assess whether FEM offers unique advantages over simpler or more interpretable methods.

3. Unclear Biological Basis for Intermediate β Values Producing Fit Sequences

The role of β as an inverse temperature parameter to sample rare mutations is reasonable. However, the non-monotonic behavior shown in Figure 2C—where intermediate β values (e.g., 0.6) yield fitter genotypes—requires stronger biological justification or empirical support. This is a key claim and must be substantiated.

4. Omission of {ATV, RTV} Combination from Critical Mutation Analysis

The exclusion of the {ATV, RTV} regimen from the critical mutation analysis (Section 2.5, page 6), despite its clinical relevance and interesting resistance profile (Figure 5), seems premature. A deeper exploration—even qualitative—may provide valuable insight into why this combination shows minimal resistance evolution.

5. Subtype Limitations and Generalizability Not Fully Addressed

All simulations are based on the subtype B consensus protease sequence. Given global subtype diversity, the generalizability of the model’s predictions, especially the identified critical mutations (Figure 6), must be more explicitly discussed in the main text.

Minor Comments:

6. Use standard mutation nomenclature (e.g., "L90M") throughout figures and text for consistency and clarity.

7. Clarify the interpretation of PCA axes in Figures 1A–B — do they represent meaningful biological separation or are they primarily for visualization?

8. Include a reference or explanation for the β-weighted sampling scheme, particularly if inspired by Boltzmann-like approaches.

Reviewer #3: Eq. 11 defined for binary sequences. However, the authors applied this equation to one-hot encoded protein sequences. One immediate consequence is that the indicator functions for different amino acids at the same positions are no longer independent, as there can be exactly one AA takes the value of 1, every other AAs must be zero. However, in Eq. 19, the authors treat the AAs at a position as if they are independent. I feel this is somewhat unnatural. So i think the authors should address this limitation.

The conditional probability for a particular AA at a position is determined by H, which is a weighted sum of contributions from AAs at other positions. This is similar to the notion of global epistasis in the evolutionary biology literature. Although this types of epistasis can capture epistasis to certain extent, it cannot account for specific interactions between amino acids. I think given the limited amount of data, this is a perfectly legitimate modeling choice. But I also think the authors should discuss the types of epistasis their model can capture. For example, can we expect the model to pick up specific pairwise interactions between residues in contact?

Overall, I feel the results section can benefit from more explanation of the method, which may be unfamiliar to many potential readers. So I suggest the authors move some material from the Methods section to the results and provide some intuitive explanation.

Also, I feel that the results sections can benefit from some motivation, as it was not immediately clear to me why the authors were performing these analyses.

**Have the authors made all data and (if applicable) computational code underlying the findings in their manuscript fully available?**

Reviewer #1: **No:** Unless I'm missing something, I can't see the codebase as part of the submission

Reviewer #2: Yes

Reviewer #3: Yes

PLOS authors have the option to publish the peer review history of their article (what does this mean?). If published, this will include your full peer review and any attached files.

Reviewer #1: **Yes:** Iain Johnston

Reviewer #2: **Yes:** Indrani Choudhuri

Reviewer #3: No

**Figure resubmission:**
---

## [Decision Letter · Decision Letter 1]

12 Nov 2025

PCOMPBIOL-D-25-00787R1

FORECASTING DRUG RESISTANT HIV PROTEASE EVOLUTION

PLOS Computational Biology

Dear Dr. Aggarwal,

Thank you for submitting your manuscript to PLOS Computational Biology. After careful consideration, we feel that it has merit but does not fully meet PLOS Computational Biology's publication criteria as it currently stands. Therefore, we invite you to submit a revised version of the manuscript that addresses the points raised during the review process.

We look forward to receiving your revised manuscript.

Kind regards,

Jessica M. Conway

Academic Editor

PLOS Computational Biology

Marc Birtwistle

PLOS Computational Biology

**Additional Editor Comments:**

Reviewer 1 continues to raise serious concerns about the readability and understandability of the paper even after one round of major revision. If the next version of the revised manuscript does not address such concerns we will not further consider the paper. This includes readability of figures such as fonts being too small (this is just one example of many).

Please also make your code publicly available in addition to your data, in line with PLoS policy. Details available at https://journals.plos.org/ploscompbiol/s/code-availability.

**Reviewers' comments:**

Reviewer's Responses to Questions

**Comments to the Authors:**

Reviewer #1: This was a confusing submission to review. Effectively the entire paper has been rewritten, but some of the more important points I raised in my review haven't been addressed: isn't the state space of interactions too big for meaning fitting? why is beta interesting?

I continue to believe that there is interesting content here. But it is super hard to pull out from the manuscript. Having made some comments previously about the difficulty of reading and intepreting this manuscript, I cannot really see that things have much improved. There are 65 figure panels to digest; many of them have illegibly small text. The core ideas are very hard to connect upon reading and the mysteries above make this harder. Fig 2 and Fig 3 appear to have been swapped around. Basically -- this is a very hard paper to read. The overall message (I think) is -- we look statistically at interactions between residues and show that this motivates simulations that inform us about drug resistance. But that message gets lost at almost every stage -- it is very hard to see how the output of an individual subsection gets us towards the goal (until the end).

I suggest that the authors focus the story on Fig 3A, Fig 2A, Fig 5*, Fig 6, Fig 7*, Fig 9*, where the * means to focus on one particular drug as the lead message then illustrate variability around that. Figs 1 and 4 provide nothing to the narrative (and unless I am misunderstanding Fig 4 and Sec 2.5 are not very interesting anyway); Fig 8 is less interesting than its neighbours, and the complexity of the article needs reducing. Other content could go to the SI. The methods need to come before the results.

l12 "are the least drug resistant" -- but you're talking about regimens here, right? "produce the least drug resistance"?

l65 -- "coevolutionary information" could do with a definition here

l70 -- the point of simulating these trajectories isn't immediately apparent here

First of all, it is essential that the methods section comes before the results section. The results section assumes a lot of knowledge about the simulations, labelling, etc that is only mentioned in the methods (there aren't even any "see Methods") links.

l88 -- isn't this an impossibly large parameter space?

The authors addressed my previous point about FEM -- but not this one (which is arguably more important). Isn't the space of this set of interactions completely huge, and much too large to be spanned by a training dataset? In fig 3 top panels for example, it looks like there must be thousands (or more) of pairs of amino acids. Aren't we dramatically overfitting?

l91 -- are the observed sequences you're comparing part of the training data, or independent? If training, see overfitting question.

l97 an improvement of 5% to 10% is a doubling of performance; 36% to 47% is only increasing by a third. Why "only" on l97?

l102 -- spatial distance ... in the sequence -- presumably this means in the folded form of the protein? As the distances between a.a. labels in the figure (which is Fig 3, not Fig 2) don't link to the distance axis.

l105 -- spatial[ly] closer

l106 -- the boxplots are different -- what boxplots? how does this link to the narrative in the text?

l129-130 -- consistent with -- in what sense? Drug resistance and mutation propensity are not the same, not even necessarily indirectly. This needs unpacking more.

l125, 132 -- Fig 3 -> Fig 2

Sec 2.5 -- as before, it seems that the result here is that higher temperature gives a more random search and lower temperatures give more canalised dynamics. The authors replied to my previous point about the biological relevance of beta by reiterating its algorithmic relevance as a noise parameter (inverse temperature). But without a connection to biology, why are the results obtained as a function of beta interesting? More random searches will always explore more space; less random searches will always be more constrained. Perhaps I am missing something here but I can't see what is unexpected either from a biological perspective ("more mutations explore more mutational space") or a physical ("more entropy explores more state space") picture? Unless I am missing something, this section and Fig 4 can basically be removed

Isn't Sec 2.6 mostly a methods section?

The story in Secs 2.7 and 2.8 is interesting, as before (2.8 needs a better title).

l278 no verb in sentence

l278, 287, 315 -- can you give permanent DOIs rather than institutional URLs?

l288 -- you gave most of these abbreviations in the previous paragraph -- just give the full set there and avoid repeating?

l304 citation for sklearn?

l312,140 is the beta superscript a power? it's ambiguous especially given the F subscript -- it would be clearer to put the probability in brackets and raise the whole thing to the power beta

Figure comments:

Why do we need figure 1? There are 22 panels and I cannot see that they provide more information than the distance statistics that are in the text.

Why do we need Fig 4? More noise makes search more exploratory? As in my comment above I am not sure this message needs a figure (or mentioning at all)

Consider including summary sentences describing the science, not the detail, for each figure, e.g.

Fig 2 -- if two residues have a high inferred interaction parameter, they are likely to be close in 3D space.

Fig 3 -- the more drug treatments an isolate is exposed to, the lower mutational variability we see at the various loci under study (lower mutation propensity)

Fig 1 -- all text is illegibly small

Fig 2 -- caption describes scatter plots, but these are boxplors. There isn't physical distance in this plot, which is part of the key message that this figure is supposed to show.

Fig 3 --

- labels (points and axes) in top panels are illegibly small

- the "None" p-value in the bottom panel surely can't be zero?

Figs 5, 7, 8 -- legend text illegibly small -- why not have a single caption for all frames and make it bigger? 8's axis text is also illegible

Fig 6 -- (B) bottom right point labels are illegibly small (and presumably not super interesting at such high precision)

IJ

Reviewer #2: The authors have satisfactorily addressed all the reviewer comments. The manuscript is now suitable for publication without further revision.

Reviewer #3: I thank the authors for making the revisions accommodating my comments. I recommend it for publication.

**Have the authors made all data and (if applicable) computational code underlying the findings in their manuscript fully available?**

Reviewer #1: **No:** I can't see a link to a codebase but I may have missed this?

Reviewer #2: **No:** The author need to provide the code and data associated with this manuscript.

Reviewer #3: Yes

PLOS authors have the option to publish the peer review history of their article (what does this mean?). If published, this will include your full peer review and any attached files.

Reviewer #1: **Yes:** Iain Johnston

Reviewer #2: No

Reviewer #3: No

**Figure resubmission:**
---

## [Decision Letter · Decision Letter 2]

8 Jan 2026

PCOMPBIOL-D-25-00787R2

FORECASTING DRUG RESISTANT HIV PROTEASE EVOLUTION

PLOS Computational Biology

Dear Dr. Aggarwal,

Thank you for re-submitting your manuscript to PLOS Computational Biology, and for being so quick with revisions.

After careful consideration, we feel that it has merit but does not fully meet PLOS Computational Biology's publication criteria as it currently stands. Therefore, we invite you to submit a revised version of the manuscript that addresses the remaining, minor points raised during the review process.

We look forward to receiving your revised manuscript.

Kind regards,

Jessica M. Conway

Academic Editor

PLOS Computational Biology

Marc Birtwistle

Section Editor

PLOS Computational Biology

**Reviewers' comments:**

Reviewer's Responses to Questions

**Comments to the Authors:**

Reviewer #1: I found this version much easier to digest and understand, and think that this version gets the key messages across in a much more accessible way. There are, to my eyes, just some small (but important) points to further explain to the reader what is going on. They are below, with particular importance assigned to a more detailed description of the training-test split and the number of parameters and datapoints.

IJ

l110 -- the "~ i" (not i) notation needs introducing here

l113 "overfitting" -- it would really help statistical intuition here to have: (a) the number of parameters in the base-rate-only model before any pruning; (b) the number of parameters in the interactions model before any pruning; (c) the impact of the regularisation (e.g. a distribution of the absolute parameter weights, which corresponds to but is not visible from a projection onto the x-axis in Fig 1A and 1B); (d) the number of datapoints.

l138-139 -- this is a bit of an orphan sentence and can be removed or folded in to the start of 3.1.1

l169 "unseen" and Table 3 caption "test" -- somewhere we need the details of the training/test split. How much of the data was used for each, were the sets independent or subsets of the same data, etc.

Figs 5 and 6 captions -- make sure the reader has all the necessary information to understand these plots from the caption alone. e.g. describe tau in both, describe phi(tau) in Fig 6

**Have the authors made all data and (if applicable) computational code underlying the findings in their manuscript fully available?**

Reviewer #1: Yes

PLOS authors have the option to publish the peer review history of their article (what does this mean?). If published, this will include your full peer review and any attached files.

Reviewer #1: **Yes:** Iain Johnston

**Figure resubmission:**
---

## [Editor Report · Decision Letter 3]

12 Jan 2026

Dear Dr. Aggarwal,

We are pleased to inform you that your manuscript 'FORECASTING DRUG RESISTANT HIV PROTEASE EVOLUTION' has been provisionally accepted for publication in PLOS Computational Biology.

Best regards,

Jessica M. Conway

Academic Editor

PLOS Computational Biology

Marc Birtwistle

Section Editor

PLOS Computational Biology

---

## [Editor Report · Acceptance letter]

PCOMPBIOL-D-25-00787R3

FORECASTING DRUG RESISTANT HIV PROTEASE EVOLUTION

Dear Dr Aggarwal,

I am pleased to inform you that your manuscript has been formally accepted for publication in PLOS Computational Biology. Your manuscript is now with our production department and you will be notified of the publication date in due course.

With kind regards,

Anita Estes
